# The Role of Immunometabolism in HIV-1 Pathogenicity: Links to Immune Cell Responses

**DOI:** 10.3390/v14081813

**Published:** 2022-08-18

**Authors:** Eman Teer, Nyasha C. Mukonowenzou, M. Faadiel Essop

**Affiliations:** 1Centre for Cardio-Metabolic Research in Africa (CARMA), Department of Physiological Sciences, Stellenbosch University, Stellenbosch 7600, South Africa; 2Centre for Cardio-Metabolic Research in Africa (CARMA), Division of Medical Physiology, BMRI, Faculty of Medicine and Health Sciences, Stellenbosch University, Cape Town 8000, South Africa

**Keywords:** HIV, non-AIDS-associated comorbidities, immunometabolism, inflammation, mitochondria

## Abstract

With the successful roll-out of combination antiretroviral treatment, HIV is currently managed as a chronic illness. Of note, immune activation and chronic inflammation are hallmarks of HIV-1 infection that persists even though patients are receiving treatments. Despite strong evidence linking immune activation and low-grade inflammation to HIV-1 pathogenesis, the underlying mechanisms remain less well-understood. As intracellular metabolism is emerging as a crucial factor determining the fate and activity of immune cells, this review article focuses on how links between early immune responses and metabolic reprograming may contribute to HIV pathogenicity. Here, the collective data reveal that immunometabolism plays a key role in HIV-1 pathogenesis. For example, the shift from quiescent immune cells to its activation leads to perturbed metabolic circuits that are major drivers of immune cell dysfunction and an altered phenotype. These findings suggest that immunometabolic perturbations play a key role in the onset of non-AIDS-associated comorbidities and that they represent an attractive target to develop improved diagnostic tools and novel therapeutic strategies to help blunt HIV-1 pathogenesis.

## 1. Introduction

Although most HIV-infected individuals on combination antiretroviral therapy (cART) achieve viral suppression, some still display viral persistence, residual inflammation, metabolic disturbances, and incomplete immunological recovery [1,2,3]. Such immunologic perturbations can elicit detrimental effects that include immune cell dysfunction, hypercoagulation, and tissue fibrosis/damage [4,5]. This can subsequently contribute to premature aging, organ dysfunction, and the development of non-acquired immunodeficiency syndrome (AIDS)-associated complications, such as cardiovascular diseases, cancers, and neurocognitive diseases [6]. Research efforts focusing on the emerging field of “immunometabolism”, i.e., exploring links between metabolic alterations and the immune system, are providing a novel perspective of immunity. Evidence also suggests that immunometabolism plays an important role in HIV-1 pathogenesis, where metabolic abnormalities may alter immune function to drive immune dysfunction and persistent inflammation, and thereby contribute to the development of non-AIDS-associated comorbidities [3,7]. For example, host metabolites can influence the immune response during HIV-1 infection and are key drivers of inflammation in this instance [8]. Furthermore, Serrano-Villar and colleagues [9] demonstrated that gut bacterial metabolism impacts immune recovery in HIV-infected individuals, and the data revealed that, most likely, reduced inflammation and immune recovery is a combined solution orchestrated by both the active fraction of the gut microbiota and the host.

The two main cellular processes responsible for glucose metabolism are oxidative phosphorylation (OXPHOS) and glycolysis. Mitochondrial OXPHOS is a highly effective process for energy production compared with aerobic glycolysis. The transition from a quiescent to activated state for T cells and macrophages needs metabolic pathway alteration (metabolic reprograming). Such metabolic switching occurs to regulate functional changes during the various stages of immune cell differentiation and activation [10].

Metabolic pathways in CD4^+^ T cells and macrophages also control their susceptibility to infection, the persistence of infected cells, and the establishment of latency [8], further emphasizing the detrimental effects of dysregulated immunometabolism. For this review article, we will, therefore, discuss the links between immunometabolism and HIV pathogenicity, as it is our view that such a focus should assist in the identification of novel diagnostic and therapeutic targets to eventually improve the clinical management of HIV patients.

## 2. Early Immune Activation and Links to Immunometabolism

Immune activation and inflammation are strongly linked to immune dysfunction and related morbidity and mortality in people living with HIV (PLWHIV) [11,12]. What is (are) the driver(s) of immune activation and how does this lead to HIV pathogenicity? The answer(s) to this rhetorical question still remain relatively unclear, but it likely includes persistent HIV replication, coinfections, and HIV-mediated microbial translocation [6,13]. However, knowledge gaps remain, as therapeutic approaches for, e.g., cART intensification [14], treatment of coinfections [15], and agents that promote mucosal repair in gut-associated lymphoid tissue [16] still do not completely resolve persistent immune activation and inflammation [6]. Further insights may be gained by focusing on metabolic pathways linked with the spontaneous loss of control in HIV “elite controllers” (HIV-positive individuals with relatively low viral load in the absence of cART) [17]. Here, investigators compared plasma metabolites and lipids in persistent elite and transient controllers and found that the latter (before losing control) displayed increased aerobic glycolysis, dysregulated mitochondrial activity, oxidative stress, and immunological activation [17]. These findings indicate that the initiation of metabolic reprograming is associated with immunological activation and dysfunction. HIV-1 also mainly infects CD4^+^ T cells and accumulating evidence reveals an association between T cell metabolic reprograming and HIV-1 pathogenesis [18]. For example, previous work investigating the expression of glucose transporter-1 (Glut1)—basal marker of glucose metabolism—in T cells reported an inverse correlation between the percentage of CD4^+^Glut1^+^ T cells and total CD4 T cell count (independent of treatment status) [19]. In support, others demonstrated that increased glucose utilization and glutamine metabolism are essential for HIV infectivity and replication in CD4 T cells [20]. Together, these studies demonstrate a link between altered glucose metabolism and markers of HIV disease progression, and that metabolic dysfunction persists even after cART initiation.

## 3. Microbial Translocation, Inflammation, and Metabolic Reprograming

During acute HIV infection, gastrointestinal tract damage can occur due to the depletion of gut-associated lymphocytes [21,22]. Here, up to 60% of CD4 T cells within the intestinal mucosa are affected and display viral RNA expression [23]. Why should this response occur at this unusual location? The gut mucosa is a primary storage site for memory T cells (expressing C-C chemokine receptor type 5) that display a semi-activated status and are a preferred target for HIV replication [24]. The normal gut lining transports antigens associated with microbes and nutrients to immune cells located in clusters, i.e., Peyer’s patches [25]. Most immune cells are located within such clusters and send an early warning to the immune system by identifying microbes to be eliminated. The main causes of microbial translocation are altered mucosal immunity that occurs due to the elimination of a relatively large proportion of CD4 T helper (Th) cells, particularly Th17 and memory T cells [26]. Subsequently, there is inflammation-associated damage to especially gap junctions within the epithelium. Such an inflammatory environment, together with increased reactive oxygen species (ROS), may lead to relatively low levels of HIV transcription and the release of proglycolytic/inflammatory extracellular vesicles by metabolically active CD4 T cells [27]. The activation of monocytes/macrophages (permanent and irreversible activation) may increase the demand for glucose, with its uptake facilitated by enhanced Glut1 expression. The relatively higher glycolytic flux in recruited monocytes and pro-inflammatory M1-like macrophages can result in the increased production of cytokines, such as interleukin-6 (IL-6) and tumor necrosis factor-alpha (TNFα), to fuel chronic inflammation in HIV-positive persons [27]. Such an inflammatory response and metabolic reprograming can persist in HIV-positive individuals, even after long-term virologic suppression by cART.

## 4. Metabolic Reprograming and Immune Responses during HIV-1 Infection

The two major features of metabolic reprograming in innate and adaptive immune cells during inflammation include: (1) increased glycolysis and decreased OXPHOS to secure more rapid ATP production and biosynthesis for a suitable defense response and damage repair; and (2) epigenetic reprograming by suppressed DNA methylation and enhanced histone acetylation [28]. There is evidence that metabolic pathways are directly linked to cell signaling and differentiation, which leads to different immune cell subsets adopting unique metabolic signatures specific to their state and environment. Here, activation of innate immune cells is initiated through toll-like receptors (TLR) expressed on antigen presenting cells (e.g., dendritic cells), macrophages, B cells, as well as specific T cells and nonimmune cells (e.g., fibroblasts and epithelial cells) [29]. They detect pathogen/danger-associated molecular pattern (PAMP/DAMP) or metabolite-associated danger signal (MADS) via pattern recognition receptors to facilitate downstream signaling cascades to initiate an immune response. The various TLRs can be roughly subclassified according to the PAMPs they recognize, i.e., TLR1, TLR2, and TLR6 detect lipopeptides, while TLR3, TLR7, TLR8, and TLR9 recognize nucleic acids. Moreover, TLR5 detects flagellin, while TLR4 recognizes a diverse collection of lipopolysaccharides [30]. Glutamine is the most abundant circulating amino acid and can be converted into α-ketoglutarate. Glutamine and glucose metabolism are interconnected, i.e., glutamine uptake via its transporter SLC1A5/ASCT2 is a rate-limiting step in the activation of the mTOR pathway (a crucial regulator of several intracellular functions), a key sensor of the cell energy status, that then leads to the upregulation of Glut1 [31]. Furthermore, Clerc and colleagues [31] demonstrated that glutaminolysis is the major pathway fueling the tricarboxylic acid cycle and OXPHOS in T-cell receptor-stimulated naïve and memory CD4 subsets, and is required for optimal HIV-1 infection. Under conditions of lowered glutaminolysis, α-ketoglutarate decreased the early steps in the infection process, with beneficial outcomes.

Loucif and colleagues [32] demonstrated the critical role of autophagy in ordering metabolic input, which is required to ensure protective cytotoxic CD8A T cell responses. It can provide a robust IL-21 production among antiviral CD4 T cells during HIV-1 infection. Here, IL-21 is only elevated among the naturally protected elite controllers, and the study confirmed a critical role for autophagy-dependent glutaminolysis in terms of IL-21 production in HIV-1-specific CD4 T cells [32].

Immune cells require a constant and adequate supply of energy for optimal functioning and cytokine synthesis/proliferation. Normal, healthy cells usually generate mitochondrial ATP through a combination of glycolysis and mitochondrial OXPHOS. However, for highly proliferating cells, such as activated Th cells and M1 macrophages, ATP and essential components used in biosynthesis are generated by switching from oxidative phosphorylation to aerobic glycolysis [33,34]. Here, higher Glut1 expression can increase glucose uptake and its supply for glycolysis. Fuel substrate switches also occur in other immune cell types, e.g., from OXPHOS to glucose utilization during the transition from naive to effector cells. Moreover, a switch from glycolysis (in effector cells) to fatty acid β-oxidation in long-term surviving memory cells can also occur [34] (Figure 1). Others also found that differences in HIV-1 susceptibility (naive versus more differentiated subsets) were associated with intracellular metabolic activity independent of their activation phenotype. Thus, persistent immune activation during HIV infection represents a unique example of metabolic remodeling, as immune cells attempt to meet energetic and functional demands by altering their metabolic profiles that are, in turn, regulated by distinct signaling cascades and transcriptional programs [10].

## 5. The Role of Monocytes/Macrophages in Metabolic Reprograming

Macrophages are the first line of defense against pathogens and become differentially activated in response to the microenvironment. There are two types of activation, i.e., the classic interferon-gamma (IFN-γ) pathway of M1 macrophages by Th1-type responses is a well-established feature of cellular immunity to HIV-1 infection. For the second mode of activation, cytokines, such as IL-4 and IL-13, are produced in a Th-2 type response, and macrophages become differentially activated (M2) and also play an important role in HIV-1 pathogenesis [35].

The classic activation pathway depends on lipopolysaccharide that is recognized as a potent activator of especially macrophages. It stimulates monocytes and macrophages via TLR4 and leads to increased TNFα secretion, the earliest and most potent proinflammatory cytokine [36,37]. While activated macrophages play an essential role in promoting systemic and chronic inflammation during HIV infection [38,39], macrophages with increased Glut1 expression produce relatively higher levels of TNFα, IL-6, and chemokine ligand-2 representing a hyperinflammatory state [40]. Moreover, lipopolysaccharide-mediated signaling in macrophages initiates a metabolic switch from OXPHOS to glycolysis [41]. While activated monocytes, macrophages (M1-like proinflammatory phenotype) exhibit a glycolytic signature, M2-like macrophages (responsible for producing anti-inflammatory cytokines) depend on oxidative metabolism and fatty acid β-oxidation for ATP generation (refer to Figure 1). Moreover, pro- and anti-inflammatory macrophages are characterized by specific pathways that regulate the metabolism of lipids and amino acids and affect their responses [33].

At another level, HIV-induced changes in systemic iron levels persist despite adequate cART treatment. For example, some demonstrated that altered iron status can influence HIV infection and replication and that such an infection impacts on both intracellular and systemic iron levels [42]. These observations establish a link between iron homeostasis and HIV infection, with intracellular macrophage iron handling as an example of this phenomenon [42].

## 6. Polarization of Th1 and Th2 during HIV Infection

Th1 cells stimulate type 1 immunity that is characterized by cell-mediated immunity and phagocyte-dependent inflammation. The main cytokines produced include IL-2, and TNFα. In contrast, Th2 cells produce IL-4, IL-5, IL-6, IL-9, IL-10, and IL-13 to stimulate type 2 immunity. This is characterized by relatively high antibody titers (humoral immunity) and eosinophil accumulation, together with the inhibition of phagocytic cell functioning (e.g., phagocyte-independent inflammation) [43,44]. Of note, Th2-dominated responses play a pathogenic role in both progressive systemic sclerosis and cryptogenic fibrosing alveolitis and favor a more rapid development of HIV infection towards a full-blown disease state [43]. Both environmental and genetic factors act in concert to determine Th1 or Th2 polarization.

For elite controllers, an immune response occurs with specific CD4 and CD8 T cells that control disease progression, i.e., highly active HIV-specific cytotoxic T lymphocytes. Here, CD4 T cells are not exhausted and are actively involved in antiviral immunity with IFN-γ production and other antiviral chemokines [45]. Furthermore, others described the cytokine profiles for Th1, Th2, and Th17, together with clinical and laboratory parameters of an HIV-infected patient with undetectable viral load without treatment (thus, an “elite controller”) [46]. Here, IL-6, IL-10, TNF-α, IFN-γ, and IL-17 was detected. It is likely that host-related factors may help explain the level of infection control, namely the differentiated modulation of the cellular immune response and a nonpolarized cytokine response of the Th1 and Th2 profiles [46].

## 7. Mitochondrial Dysfunction and Reprograming in HIV-Positive Patients on cART

Mitochondria are dynamic organelles that are present in most eukaryotic cells and responsible for energy production [47]. Mitochondria can also regulate the activation, differentiation, and survival of immune cells, while also releasing signals, such as mitochondrial ROS, to regulate transcription of immune cells. Mitochondrial shape and its intracellular localization are crucial for optimal functioning and are, therefore, tightly regulated by fusion and fission processes.

In the context of HIV, mitochondrial dysfunction can be mediated by both the virus and long-term cART to eventually result in cellular exhaustion, senescence, and apoptosis [48,49]. Since mitochondria are essential intracellular organelles for energy homeostasis and cellular metabolism, their dysfunction leads to decreased OXPHOS, ATP synthesis, gluconeogenesis, fatty acid β-oxidation, abnormal cell homeostasis, increased oxidative stress, depolarization of the mitochondrial membrane potential, and upregulation of mitochondrial DNA mutations and cellular apoptosis [50]. The progressive mitochondrial damage induced by HIV infection and cART treatment likely contributes to accelerated aging, senescence, and cellular dysfunction in HIV-infected persons [50]. As HIV-infected patients are more prone to premature aging, an increased mitochondrial oxidative state leading to metabolic disturbances can further exacerbate this condition [51].

HIV itself can contribute to mitochondrial dysfunction and apoptosis in CD4 and CD8 T cells [49]. The mechanisms underlying this dysfunction include depletion of mitochondrial DNA, decreased mitochondrial gene expression, reprograming of energy production via OXPHOS, and increased ROS production [48]. There is also evidence that HIV-related mitochondrial dysfunction may be linked to treatment regimens. Here, nucleotide reverse transcriptase inhibitors are mainly responsible for mitochondrial dysfunction in adipose tissue and liver, although non-nucleoside transcriptase inhibitors and/or protease inhibitors can also trigger detrimental mitochondrial sequelae [52].

## 8. Conclusions

There is strong evidence that immune dysregulation is the main driver for HIV pathogenicity. However, recent studies show that metabolic reprograming of immune cell subsets is linked to such immunological activation and dysfunction. These findings suggest that immunometabolic perturbations play a key role in this context and that it represents an attractive target to develop improved diagnostic tools and novel therapeutic strategies to help blunt HIV-1 pathogenesis and non-AIDS-associated comorbidities.

## Figures and Tables

**Figure 1 viruses-14-01813-f001:**
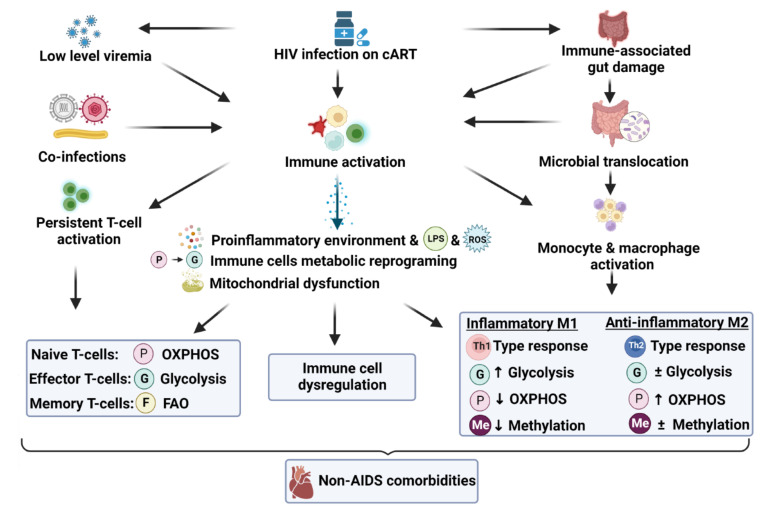
The role of immunometabolism in HIV-related immune activation and dysfunction. Abbreviations: cART: combination antiretroviral therapy; FAO: fatty acid β-oxidation; LPS: lipopolysaccharide; OXPHOS: oxidative phosphorylation (mitochondrial); ROS: reactive oxygen species; Th: T helper cells.

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
