# Peer review of "The Role of Immunometabolism in HIV-1 Pathogenicity: Links to Immune Cell Responses"

_viruses, 2022, doi:10.3390/v14081813_

Round 1

Reviewer 1 Report

The manuscript by Teer and colleagues present a review of the role immunometabolism and immunoactivation in the pathogenicity of HIV-1.  It discusses the role of HIV-1 induced damage to the gut associated lymphoid tissue GALT, persistent immunoactivation by translocation of bacteria and co-infections.  Finally, the authors discuss the role of the virus and cART on

mitochondrial dysfunction that eventually results in cellular exhaustion, senescence, and apoptosis. Overall, I found that this manuscript was well written and is an important topic in HIV-1 infections that needs to be addressed. 

Author Response

Thank you for taking the time to assess our manuscript and for your valuable contribution to our article.

Reviewer 2 Report

The manuscript entitled, The Role of Immunometabolism in HIV-1 Pathogenicity: Links to Immune Cell Responses is an excellent review. In resume, the link between HIV pathogenesis and chronic immune activation and inflammation appears to be based on the dysregulation of immune cell metabolism during HIV infection. Metabolic reprogramming of virus-activated cells leads to immune-metabolic dysregulation, which contributes to non-AIDS comorbidities. The metabolic derangements result in increased aerobic glycolysis, altered mitochondrial activity and oxidative stress.

Minor corrections:

I suggest adding in this section Metabolic reprogramming and immune responses during HIV-1 infection,  comment on this topic.

The cellular metabolic pathway that the review does not seem to mention is glutaminolysis, which involves the conversion of glutamate to É‘-ketoglutarate and provides energy for cell proliferation (https://www.frontiersin.org/articles/10.3389/fimmu.2020.01013/full), PMID: 34612140, PMID: 32373781

Originality of study

This article is original. 

Scientific quality

It is a well-written manuscript, and the figure is adequately presented.

Impact of the research

The impact of this work is excellent.

Author Response

Thank you for identifying the weaknesses in our paper and providing the opportunity to strengthen our paper prior to publication. Thank you for your correction with regards to glutaminolysis. It plays a crucial role in metabolic reprogramming and immune responses. We have now used the suggested papers to add to this topic. Please refer to the original paper in the section of Metabolic reprogramming and immune responses during HIV-1 infection (yellow color).